# Clinical Pharmacists’ Knowledge of and Attitudes toward Older Adults

**DOI:** 10.3390/pharmacy9040172

**Published:** 2021-10-20

**Authors:** Tasia Karis Allen, Patrick Mayo, Sheri Koshman, Margaret Gray, Amina Babar, Cheryl Ann Sadowski

**Affiliations:** 1Pharmacy Services, Alberta Health Services, University of Alberta Hospital, Edmonton, AB T6G 2B7, Canada; Tasia.KarisAllen@albertahealthservices.ca (T.K.A.); Margaret.Gray@albertahealthservices.ca (M.G.); 2Faculty of Pharmacy and Pharmaceutical Sciences, University of Alberta, Edmonton, AB T6G 1C9, Canada; pmayo@ualberta.ca; 3Division of Cardiology, Faculty of Medicine and Dentistry, University of Alberta, Edmonton, AB T6G 2B7, Canada; skoshman@ualberta.ca; 4Pharmacy Services, Covenant Health, Edmonton, AB T5K 0L4, Canada; amina.babar@covenanthealth.ca

**Keywords:** geriatrics, aged, attitudes, knowledge, pharmacy, ageism

## Abstract

Background: Although pharmacy literature suggests that pharmacists have a positive attitude towards older adults, there is a paucity of studies that have measured pharmacists’ knowledge or attitudes towards older people. The purpose of our study was to assess the knowledge and attitudes of hospital pharmacists toward older adults. Methods: An electronic survey was distributed over two months to clinical hospital pharmacists across the province of Alberta, Canada. The survey was composed of two validated tools, the Palmore Facts of Aging Quiz (PFAQ) and Kogan’s Attitude toward Old People Scale (KOPS). PFAQ is scored from 0 (poor knowledge) to 25 (high knowledge) and KOPS from 34 to 204, with higher than 119 indicating a positive attitude. Results: A total of 153 pharmacists completed the survey (response rate of 24%). The mean age was 39 (SD 10.8) years; the average years practiced was 15 (SD 11), and the majority of respondents (*n* = 65) reported that >50% of patients in their practice were geriatrics. The mean correct responses on the PFAQ were 18.8 (SD 2.6). KOPS had a mean score of 156.8 (SD 14), with only one pharmacist score falling below 119, indicating a negative attitude. There was a statistically significant, positive correlation between attitudes and knowledge (*r* = 0.38, *p* < 0.05), as well as the increasing age of the respondents (*r* = 0.18, *p* = 0.03). The remaining measured categories (i.e., gender, years of pharmacy practice) had no significant effect. Conclusion: Clinical hospital pharmacists in Alberta have a positive attitude toward geriatric patients, as well as a satisfactory knowledge of older adults.

## 1. Introduction

By the year 2036, the proportion of Canadian adults older than 65 is estimated to reach 25% [1]. Internationally, over half of the population of older adults experiences polypharmacy [2]. In Canada, seniors use more medications than any other age group, with 66% of seniors using five or more drugs and 27% using 10 or more [3]. Pharmacists have the expertise to address medication-related problems, which improves safety [4], decreases inappropriate drug use [5,6] and reduces overall expenses for the health system [6]. With the growth of the aging population, pharmacists choosing to practice in a geriatric setting will be an important asset for a potentially overburdened health system.

In view of increasing healthcare demands it is important for institutions to assess the knowledge and attitudes of their employees, as practitioners who show better attitudes and increased knowledge are generally more interested in working with older adults [7,8] and may provide better care to this population [9,10]. There is evidence that negative attitudes affect decision making and clinical care across all specialties and healthcare settings [11]. However, the literature shows that healthcare workers have varying attitudes [12,13,14,15]. Health professionals, like the rest of society, can have positive (e.g., being kind, generous or wise) or negative (e.g., being cranky, sick or demanding) stereotypes that affect their attitudes toward older adults [16,17].

There are concerns from organizations such as the Institute of Medicine which identified negative attitudes toward older adults as being persistent in healthcare and a challenge in preparing for demographic changes that will affect the healthcare system [18]. Additionally, many professions indicate that further training for improved geriatric care is required [19,20,21,22,23,24,25].

The published literature pertaining to pharmacists’ knowledge and attitudes is limited. One Canadian study from 1991 evaluated pharmacists [26], and there is one study from Saudi Arabia [27] and one from Malaysia [28]. There have been multiple studies of pharmacy students showing that educational interventions result in positive student attitudes [29,30,31,32,33,34,35,36,37,38]. A longitudinal study found that pharmacy students maintained positive attitudes toward older adults throughout the pharmacy curriculum [39]. Unfortunately, there is little research on the knowledge or attitudes of pharmacists in practice, particularly those in a hospital setting who care for frail and acutely ill seniors.

The aim of this study was to investigate the attitudes and knowledge of clinical hospital pharmacists toward older adults and to determine whether a relationship exists between knowledge and attitudes.

## 2. Materials and Methods

### 2.1. Objectives

#### 2.1.1. Primary Objectives

Determine the attitudes and knowledge of clinical hospital pharmacists toward older adults.

#### 2.1.2. Secondary Objectives

Determine whether there is a relationship between knowledge and attitudes and if it differs based on: gender, age, years of pharmacy practice, highest level of geriatric education and percent of contact with geriatric patients in clinical practice.

### 2.2. Study Design and Timeframe

This study was an electronic survey of clinical hospital pharmacists that practice within Alberta Health Services (AHS) (province wide) and Covenant Health (Edmonton only), in Alberta, Canada. The initial survey as well as two reminder emails were distributed between 24 February and 23 March of 2015.

### 2.3. Data Sources

Clinically deployed pharmacists were identified through the employer’s pharmacist database. Email addresses were subsequently retrieved from an internal email directory.

The survey was developed by combining two previously validated scales, “Kogan’s Attitudes toward Older People Scale” (KOPS) [40] and “Palmore Facts on Aging Quiz: Canadian Version” [41]. Demographic questions to assess the population and secondary objectives were developed, including age, gender, years of pharmacy practice, highest level of geriatric education and percent of time in contact with geriatric patients in clinical practice (see Appendix A).

#### 2.3.1. Palmore Facts on Aging Quiz

PFAQ was created by Erdman Palmore in 1977 [42] and was assessed in a Canadian population to ensure cross-cultural validity [41]. The questionnaire was designed with 25 true/false questions to test factual statements that cover basic physical, mental, social facts, and the common misconceptions about aging. The mean score is used to assess the general knowledge that pharmacists have of older adults. Another use of PFAQ can be to indirectly measure the bias toward the aged. By measuring the mean percentage of errors on questions (See Appendix A) that indicate a negative bias (questions 6, 8, 10, 12–16, 18, 21), as well as those that indicate a positive bias (question 7, 9, 11, 17, and 19), the net anti- or pro-aged score is calculated by subtracting the mean percent errors on negative bias from the positive. A negative score indicates a net anti-aged bias, and a positive score is a pro-aged bias.

A review of PFAQ was done by comparison with Statistics Canada demographics, and as of July 2014 the question, “over 15% of the Canadian population are now age 65 or older”, is currently true instead of false [1]. An additional question, “the health and socio-economic status of older people (compared to younger people) in the year 2000 will probably be about the same as now”, has been modified to an adjusted date of 2037. The correct answer for this question remains false [43].

#### 2.3.2. Kogan’s Attitude toward Old People Scale

KOPS was developed in 1961, and since then its development has been validated and acknowledged as an accepted tool [40,44,45] and applied across a range of groups, including various health professionals [17,23,44,45,46,47,48]. The scale consists of 34 items, separated into 17 positive and 17 negative statements. Each statement is measured on a 6-point Likert scale. Each positive statement is scored by forced choice as follows. 6 points: strongly agree, 5 points: agree, 4 points: slightly agree, 3 points: slightly disagree, 2 points: disagree, and 1 point: strongly disagree; the reverse scoring system is used for the negative statements.

KOPS scores range from a minimum of 34 to a maximum of 204, with a higher value indicating a more positive attitude. Since KOPS had no validated ranges or set points that defined a ‘good’ or ‘poor’ attitude, a midpoint neutral attitude value was determined to be 119 [26,46]. A total score above the midpoint was considered a relatively positive attitude and one below was considered a relatively negative attitude.

### 2.4. Inclusion and Exclusion Criteria

Clinical pharmacists that worked within Pharmacy Services at AHS and Covenant Health, who were deployed to direct patient care (as defined by the pharmacist database, which excludes casual and dispensary only pharmacists) for at least a portion of their role, were eligible for inclusion. This ensured that the surveyed pharmacists were those providing direct patient care and that they were consistently available during the survey period, ensuring a greater chance of subject participation. Those pharmacists identified through the database whose emails were not found or returned undelivered were excluded from the study.

### 2.5. Data Collection

Requests for survey completion were distributed through AHS and Covenant Health email addresses. The email distributed to pharmacists included the information letter and a link to electronically complete the survey. The participants were informed of the anonymity of the survey and that the submission of responses would imply consent. Once information was submitted, the participant was unable to withdraw consent due to the anonymity. The electronic survey was collected through a Google Forms survey [49].

Participants were included in the analysis if they completed the demographic information and either the KOPS or PFAQ or both. Each questionnaire was required to be fully completed in order to be able to calculate a score, to reflect how the tools were validated.

### 2.6. Statistical Analysis

Descriptive statistics were calculated based on the KOPS and PFAQ total scores. The results were analyzed using R v3.02 [50]. Spearman’s rank correlation was used to analyze the relationship between knowledge (PFAQ) and attitude (KOPS) scores, gender, age, the percentage of contact with geriatrics in current practice, the years of pharmacy practice experience and the highest level of geriatric education. The normality of the data was tested using Shapiro–Wilks. An ANOVA two-way analysis (significance *p* < 0.05) and Tukey’s range test were used to compare differences in knowledge and attitudes, the percentage of contact with geriatrics in current practice, the years of pharmacy practice experience and the highest level of geriatric education.

### 2.7. Ethics Approval

Ethics approval was obtained through the University of Alberta Health Research Ethics Board.

## 3. Results

A total of 663 clinically deployed pharmacists were identified within the pharmacist database. Of these, 31 were excluded and 153 (24%) completed the survey, as shown in Figure 1. The characteristics of the pharmacists are summarized in Table 1. The mean age of pharmacists was 39 years, most were female (82%), and >50% reported that contact with geriatric patients in their clinical practice was common (43%). Most commonly, pharmacists stated that their highest level of education regarding geriatrics was practice-based educational experience with this population.

### 3.1. Knowledge Regarding Older Adults

The results of the PFAQ and KOPS are summarized in Table 2. PFAQ had a mean correct answer of 18.8 +/− 2.6, (which equals 75.2% +/− 10.5), indicating a high level of knowledge. An analysis of variance for years of practice experience, percent of contact with geriatric patients and highest level of education did not demonstrate a statistically significant effect on knowledge.

An examination of the incorrect responses to questions revealed that 24% of the questions had an error rate greater than 40%, and 12% had an error rate higher than 50%. The most common misconceptions included the following: that medical practitioners do not give low priority to the aged, the majority of old people are often bored and at least one-tenth of the aged are living in long-stay institutions. More pharmacists appeared to correctly answer physical and mental questions, as classified by Palmore. This included ‘the majority of old people (past age 65) are not senile’, ‘most old people are interested in, or have the capacity for, sexual relations’, and ‘it is possible for old people to learn new things’.

The indirect measure of the response bias was computed by examining the pro-aging score (20.9) in comparison to the anti-aging score (24.7). The net score of −3.8 showed that pharmacists made slightly more anti-aging errors than pro-aging errors; however, there was relatively little difference between the two scores.

### 3.2. Attitudes toward Older Adults

Pharmacists had an overall positive attitude, as indicated by the KOPS total mean score of 156.8 +/− 13.9. Only one pharmacist score fell below the neutral value of 119, indicating a negative attitude. The respondent age showed a statistically significant but low correlation with attitudes (r = 0.18, *p* = 0.03), as shown in Table 3. An analysis of variance for years of practice experience, percent of contact with geriatric patients and highest level of education did not demonstrate a statistically significant effect on attitude.

### 3.3. Correlation between Knowledge and Attitudes

As shown in Table 3, the Spearman correlation showed a statistically significant, positive correlation (*r* = 0.38, *p* < 0.001) between knowledge and attitudes.

## 4. Discussion

Overall, the findings of this study indicate that clinical pharmacists practicing in a hospital setting have positive attitudes and demonstrate a good level of general knowledge of older adults. A further comparison suggested that a higher level of knowledge correlated with more positive attitudes.

These findings are consistent with a previous Canadian study, which also evaluated pharmacists’ attitudes using KOPS. Their study concluded that 95% of the pharmacists practicing in the community and hospitals had a positive attitude toward geriatric patients [26], which parallels our study where only one pharmacist was below the neutral value. While our study showed a weak correlation between attitude and participant age, their study found no association based on demographic and practice variables. Other international studies found that a medication game improved empathy and increased knowledge for pharmacists in Saudi Arabia [27], and a Malaysian training program developed for community pharmacists increased knowledge and comfort with care [28], but neither study directly evaluated attitudes.

This trend of positive attitudes may be affected by the foundation that may be laid for pharmacy students who complete a focused educational activity on geriatrics. A variety of educational activities have been studied, including entire courses, workshops, simulations, gaming, health screening clinics, film studies, virtual assessments, structured interaction with seniors, and geriatrics-based rotations (e.g., long-term care facilities) [29,30,31,32,33,34,35,36,37,38]. These studies generally involve a pre/post assessment without a control group, although there was one comparison of students who had already completed a course versus those who had not yet completed the course [31] and one service learning intervention that had a control group [34]. There may be a publication bias in the studies involving students, or it may be that any form of intervention of exposing pharmacy students to older adults leads to positive attitudes.

Other studies demonstrate a variation in attitudes within professions, with nurses and physicians showing negative, neutral, and positive attitudes in systematic and integrative reviews [13,14,15,51,52]. It is difficult to compare these subjects across different studies, in part because a variety of scales have been used, with some being tailored for only one healthcare profession. However, the evidence seems to suggest that positive attitudes by pharmacists may be better and more consistent than the other health professions studied [13,14]. It is unclear why this is, but there may be multiple potential explanations. One potential reason may involve the practice environment. For example, pharmacists focus on medications, while nurses provide broader care, with more physical contact, such as assisting with soiled diapers, inserting catheters, or in a situation where behavioral problems could result in physical harm [52]. This may contribute to their perception and make caring for frail seniors less desirable. Additionally, the culture of pharmacy may be more senior-friendly, with pharmacists having a significant presence serving seniors in the community. Since most pharmacists practice in a community setting, they may see mostly healthy or fit seniors, versus the complex and frail seniors in long-term or acute care. However, even in our sample of mainly acute care pharmacists, the attitudes were positive. The drivers for this require further investigation, as this information could be applied to an intervention or selection of other health professionals.

Unlike attitudes, the PFAQ score allows for a comparison of pharmacist general knowledge to that of other healthcare professionals. Pharmacists in our study appear to have a slightly higher general knowledge in comparison to occupational therapists at 66%, physicians at 68%, and physical therapists at 70% [16,53].

Attitudes are complex and may be developed at a very young age, which may explain why variables, such as the age, gender and education level, were not consistent predictors of attitudes across a nursing population [14]. Even the correlation between knowledge and attitudes is controversial. Our study supported other nursing studies [14,23] that showed a positive correlation. However, a systematic review of medical students and physicians found that increased knowledge did not affect attitude but that an empathy-building task in an intervention was associated with positive attitude change [15,54].

The strengths of our study include a province-wide distribution, making the results more generalizable to the hospital pharmacist population. We focused on one of the least studied professions, yet one that has a high frequency of interaction with and time spent caring for older adults. We used validated tools and ensured that they were corrected to current demographics and statistics. We studied both knowledge and attitudes. Limitations include a response bias since only 24% of pharmacists participated in the study, even though this is expected in an online survey [55,56]. Second, when studying attitudes, participants might feel increased pressure to provide a socially acceptable answer. To help mitigate this pressure, the surveys were submitted anonymously to increase the likelihood of an honest response.

## 5. Conclusions

Given the limited data regarding pharmacist knowledge and attitudes and the growing Canadian geriatric population, our results have the potential to impact care positively as the attitudes of healthcare professionals have been shown to affect the quality of care provided [10,11,12]. Pharmacists having a positive attitude and advancing scope of practice are thus ideal professionals to enhance geriatric care and to lead by example with other professions that have poorer attitudes. These findings can direct administrators to engage pharmacists to help alleviate healthcare demands by selecting pharmacists based on positive attitudes in addition to the usual skills and attributes, assigning and supporting them to geriatrics care settings, adequately resourcing those work settings, and placing them in leadership roles amongst other pharmacists, with trainees, and in interprofessional teams, where they can impact others [52].

The findings offer support for previous research that pharmacists have a positive attitude toward older adults. It also provides new data indicating that they have a satisfactory general knowledge of older adults. The weak relationship between knowledge and attitudes and the minimal influence from the demographic characteristics show that attitude is affected by additional factors. Further research is required to assess the factors that correlate with attitude and to address pharmacist-specific geriatric medication knowledge.

## Figures and Tables

**Figure 1 pharmacy-09-00172-f001:**
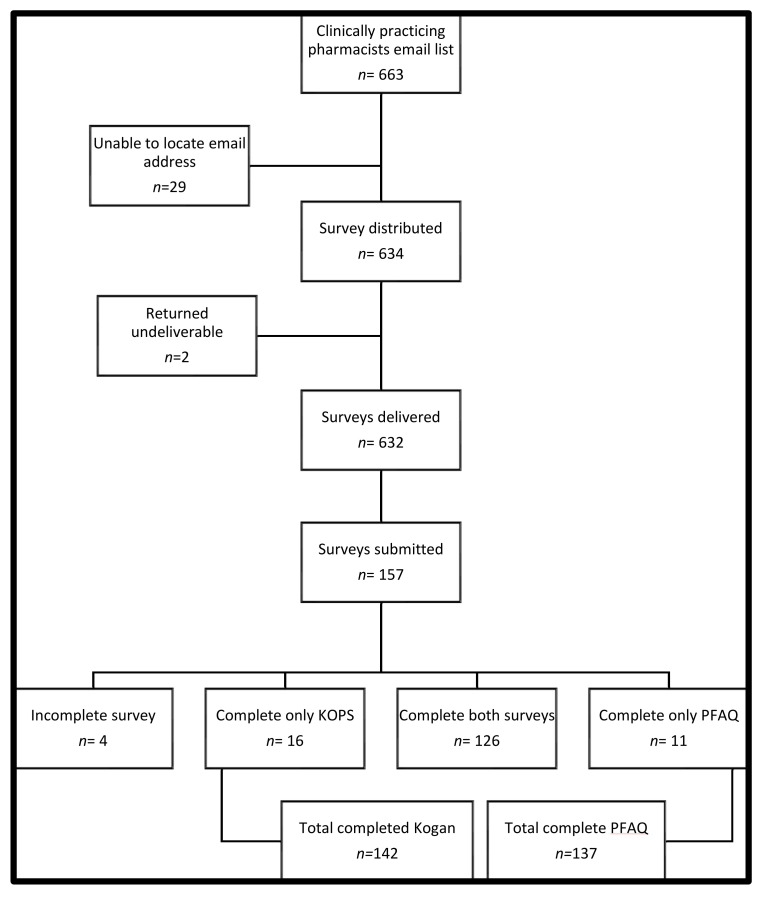
Collection of surveys for hospital pharmacists.

**Table 1 pharmacy-09-00172-t001:** Characteristics of responding clinical pharmacists.

Characteristic	Results
**Total Population *n* (%)**	153 (24)
Female *n* (%)	126 (82)
Age mean (SD)	39 (10.8)
Years of practice experience mean (SD)	15 (11.1)
**Contact with geriatrics patients in clinical practice *n* (%)**	
None	8 (5)
Less than 25%	23 (15)
Between 25–50%	57 (37)
Greater than 50%	65 (43)
**Highest level of geriatric education *n* (%)**	
Bachelor’s Degree	35 (22.9)
Continuing Education Course	13 (8.5)
Practice Related Experience	72 (47)
Residency Program or Doctor of Pharmacy	26 (17)
Certified Geriatric Pharmacist	7 (4.6)

**Table 2 pharmacy-09-00172-t002:** Kogan and Palmore total scores based on pharmacist characteristics.

Characteristic	Kogan Total Score	Palmore Total Score
Gender		
Female	157.1 (14.2)	75.6 (10.6)
Male	155.9 (13.5)	73.7 (10.3)
Percent of Contact with Geriatric Patients in Current Practice		
None	156.8 (11.9)	79.4 (5.38)
Less than 25%	153.4 (15.2)	71.1 (12.1)
25–50%	156.2 (13.9)	74.7 (10.7)
Greater than 50%	158.7 (13.9)	76.7 (9.9)
Highest Level of Geriatric Education		
Undergraduate in Pharmacy	154.0 (14.9)	73.7 (10.6)
Continuing Education Course	155.1 (14.2)	70.5 (15.4)
Practice Experience	156.8 (13.2)	76.4 (10.0)
Residency/Doctor of Pharmacy	160.8 (14.3)	75.5 (7.8)
Certified Geriatric Pharmacist	159.8 (15.9)	81.3 (10.7)

**Table 3 pharmacy-09-00172-t003:** Comparison of attitudes and knowledge using the Spearman correlation.

Score	Comparator	Correlation *r*	*p*-Value
Attitude	Knowledge	0.38	*p* < 0.0001
	Age	0.18	*p* = 0.03
	Gender	−0.03	*p* = 0.72
	Years of Practice Experience	0.16	*p* = 0.07
	Percent of contact with geriatric patients in current practice	0.08	*p* = 0.32
	Highest level of education	0.16	*p* = 0.053
Knowledge	Age	0.17	*p* = 0.054
	Gender	−0.06	*p* = 0.47
	Years of Practice Experience	0.12	*p* = 0.15
	Percent of contact time with geriatric patients	0.08	*p* = 0.35
	Highest level of education	0.1	*p* = 0.23

## Data Availability

Not Applicable.

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
