# Peer review of "Clinical Pharmacists’ Knowledge of and Attitudes toward Older Adults"

_pharmacy, 2021, doi:10.3390/pharmacy9040172_

Round 1
Reviewer 1 Report
Dear authors,
This is a neatly written manuscript that solves an important problem.
I hope you correct a few points.
1. Please add more references on the role of pharmacists in supporting elderly patients in other countries to emphasize the role of our profession in managing an aging population, not only in Canada but globally. 2. Secondly, I missed the conclusion part. Please separate it from the discussion. 3. Please remove this sentence on line 202, I believe it should be there.
Thank you.

Author Response
Thank you for the comments and review. We have made the following changes, in addition to suggested changes from reviewer 2:
- 2 additional references were added for international pharmacy practice and attitudes toward older adults
- additional citations were added for both American and EU perspectives
- line 202 was removed
Reviewer 2 Report
Clearly this paper represents an under-studied area that is increasingly relevant given the rising size of the senior adult population in Canada and many other countries around the world. The authors cite many studies of knowledge and attitudes toward the elderly in other disciplines but apparently found little in their literature search specifically on pharmacists' K&A. I am somewhat surprised that their survey was conducted in 2014-15 and only one citation is newer than 2016. Has there been additional work in any health field but specifically pharmacy published more recently?
I note some fairly obvious edits are needed but would note that their methods are clearly described and appropriate, including the use with some appropriate adaptation of validated instruments used in other relevant fields. Their figures and tables are clear and informative.
On line 202 of the manuscript there is a partial sentence that appears to be a recommendation to the authors. I assume that this has been addressed but clearly that should be deleted. I am uploading the copy of the paper I used which has some areas where edits are needed or might be considered (word choices).
The authors appear to have blended the Discussion section and a missing Conclusion paragraph or two. I think they could break up the last couple of paragraphs into an adequate Conclusion but this is a deficiency in this draft.
Notwithstanding the lag between data collection and publication, this manuscript can be improved with a bit of additional effort by the authors and published.

Author Response
We would like to thank reviewer 2 for the detailed and helpful review.
We have updated the Introduction and Discussion to cite more recent research in this field.
Grammar, punctuation, and wording were addressed as suggested on the scanned document.
The Conclusion has been separated from the Discussion.